# Comparing actuarial and subjective healthy life expectancy estimates: A cross-sectional survey among the general population in Hungary

Zsombor Zrubka[1,2]*, Áron Kincses[3,4], Tamás Ferenci[1], Levente Kovács[1], László Gulácsi[1,2], Márta Péntek[1]

1 University Research and Innovation Center, Óbuda University, Budapest, Hungary, 2 Corvinus Institute of Advanced Studies, Corvinus University of Budapest, Budapest, Hungary, 3 Hungarian Central Statistical Office, Budapest, Hungary, 4 Institute of World and Regional Economics, University of Miskolc, Miskolc-Egyetemváros, Hungary

* zrubka.zsombor@uni-obuda.hu

**Data Availability Statement:** All relevant data are within the manuscript and its Supporting Information files.

## Abstract

### Background

Healthy life expectancy (HLE) is becoming an important indicator of population health. While actuarial estimates of HLE are frequently studied, there is scarcity of research on the subjective expectations of people about their HLE. The objective of this study is to compare actuarial and subjective HLE (sHLE) estimates in the ≥50-year-old Hungarian general population. Furthermore, we assessed subjective life expectancy (sLE) and explored determinants of the individual variance of sHLE and sLE.

### Methods

We conducted a cross-sectional online survey in 2019. Subjective health expectations were measured at 60, 70, 80 and 90 years of age via the Global Activity Limitation Indicator (GALI). Point-estimates of sLE and background variables were also recorded. sHLE was estimated from GALI and sLE responses. Actuarial estimates of life expectancy (LE) and HLE for 2019 were provided by the Central Statistical Office of Hungary.

### Results

Five hundred and four respondents (female 51.6%) were included. Mean (±SD) age was 63 (±7.5) years. Median LE (81.5 years, 95%CI 81.1–81.7) and sLE (82 years, 95%CI 80–85) were similar ($p = 0.142$), while median sHLE (66.8 years, 95%CI 65.5–68.3) was lower than HLE (72.7 years, 95%CI 82.4–82.9) by 5.9 years ($p<0.001$). Despite the greater median actuarial LE of women compared to men ($p<0.001$), we found no gender differences between the median sLE ($p = 0.930$), HLE ($p = 0.417$) and sHLE ($p = 0.403$) values. With less apparent gender differences among the predictors when compared to sLE, sHLE was mainly determined by self-perceived health, age and place of residence, while self-

**Funding:** Data collection was supported by the Higher Education Institutional Excellence Program of the Ministry of Human Capacities in the framework of the 'Financial and Public Services' research project (20764-3/2018/FEKUTSTRAT) at Corvinus University of Budapest. The conduct of this study was supported by the Higher Education Institutional Excellence Program of the Ministry for Innovation and Technology in the framework of the "Thematic Excellence Program" research project (TKP2020-NKA-02) at the Corvinus University of Budapest. Financial interests: In connection with writing this article, ZZ, ÁK, TF, LK, MP and LG received grant support from the Higher Education Institutional Excellence Program of the Ministry for Innovation and Technology "Thematic Excellence Program" (TKP2020-NKA-02) at the Corvinus University of Budapest. Non-financial interests: MP is member of the EuroQol Group, a not-for-profit organisation that develops and distributes instruments that assess and value health.

**Competing interests:** The authors have declared that no competing interests exist.

perceived health, close relatives' longevity, social conditions, happiness and perceived lifestyle influenced sLE.

## Conclusions

Along subjective life expectancy, subjective healthy life expectancy may be a feasible indicator and provide insights to individuals' subjective expectations underlying the demographic estimates of the healthy life expectancy of the population.

## Introduction

Since the second half of the 20th century, average life expectancy (LE) has been rising and the share of the elderly in the total population has been growing steadily. LE at birth in the most developed countries has reached 80 years. While LE at birth has increased globally by 5.5 years between 2000 and 2016, in the same period, health adjusted life expectancy (HALE) at birth has increased by 4.8 years suggesting that despite the increased healthy life span, people also live longer with disabilities [1]. Hence, healthy life expectancy (HLE) has been gaining importance and has been used for planning and evaluation of health policies [2,3]. The importance of health-related quality of life is emphasized by the World Health Organization's (WHO) European Health 2020 policy framework [4]. Furthermore, the Lisbon Strategy of the European Union includes the target of adding on average two healthy life years across the EU by 2020—a target that still needs to be met [5].

WHO and Eurostat compute HLE via the Sullivan-method using standard life tables and cross-sectional gender- and age-specific morbidity data. For each corresponding age group in the life table, the proportion of time with disability is subtracted, and future years spent without disability are summarized [6–8]. While WHO computes HALE using disability estimates from the Global Burden of Disease Study [8], Eurostat's Healthy Life Years (HLY) are based on the Global Activity Limitations Indicator (GALI) collected regularly by all European Member States in the European Union Statistics on Income and Living Conditions survey (EU-SILC) [6,9]. The GALI comprises a single question to assess long-standing activity limitations due to health problems [10,11]. Hence, the European HLY is also referred to as Disability Free Life Expectancy (DFLE) [6]. The Hungarian Central Statistical Office (HCSO) publishes HLY along with LE indicators annually [12].

While LE estimates provide information on the general status of the population, subjective life expectancy (sLE) has been studied as a proxy to gain insight into individual variances of LE [13]. The determinants of sLE include the longevity and health of forebears [14–18], health status [16,18–20], age [16,18,19], gender [16,18,21–23], lifestyle-related risks [18,24–26], socio-economic status [21,27] as well as a number of psychosocial and psychological factors, such as happiness, optimism, social relationships, depression or the sense of control [18,20,28–31]. sLE has proven to be a predictor of actual life-expectancy and mortality [19,32,33], a determinant of decisions about retirement [34,35], consumption and savings [36] as well as health behaviours [37,38]. The deviation of sLE from actuarial LE has been associated with potentially important economic consequences due to altered perceptions on one's financial prospects [14,26]. Furthermore, sLE influences the subjective value people attach to different levels of health problems [39,40].

Considering the breadth of research concerning sLE, there is scarcity of research on subjective healthy life expectancy (sHLE). However, subjective predictions about the onset and

severity of disability might have also important implications for individuals' current health-related and financial decisions as well as personal old age planning. These potential effects are of special importance if individuals' ideas regarding future health (including both longevity and health problems) reflect an over- or underestimation compared to what can realistically be expected. Previous research on sHLE in the Netherlands [41,42] and Hungary [43,44] suggested that people underestimate their future health. However, these studies gauged health-related expectations via EQ-5D-3L, a generic health-related quality of life instrument [45,46], so their results are not directly comparable with Eurostat's HLYs obtained via the GALI.

Therefore, our aim was to compare subjective and actuarial estimates of healthy life expectancy in the ≥50-year-old general population of Hungary, by adapting the GALI instrument to assess future health expectations. Furthermore, we assessed sLE and explored the determinants of the individual variance of sHLE and sLE.

## Methods

### Sample and study design

This study was performed on the ≥50-year-old subsample (study sample) of a larger cross-sectional online survey of health- and longevity-related expectations conducted among the ≥18-year-old general population of Hungary in 2019 (survey sample) [20]. Ethical approval for the study was obtained from the Medical Research Council of Hungary (5113-2/2018/EKU). Participation was anonymous and respondents gave written consent prior completion of the survey. The survey sample comprised 1000 participants recruited from an online panel by a market-survey company using sampling quotas proportional to the general adult population in terms of gender, age (in the 18-65-year-old age group), educational level, type of settlement and geographical region. Those respondents were selected in the study sample, who were ≥50-year-old and provided coherent answers to questions related to future health expectations from 60 years of age. Younger individuals were excluded to avoid inflated measurement error when the onset of expected disability was ≤60 years.

### Subjective and actuarial estimates of life expectancy and healthy life expectancy

We inquired sLE by asking a subjective point-estimate from each respondent [47] (Section A in S1 Appendix). Although survey answers were accepted without restriction, we included in our analysis respondents if their answers fell between their own age and the upper limit of 100 years.

The terms healthy, without limitation, or without disability will be used interchangeably throughout this paper. Future health expectations were measured with the adapted GALI [10], by asking the level of expected limitations due to health problems at the age of 60, 70, 80 and 90 years, whichever was higher than respondents' age (Section B in S1 Appendix). GALI was also administered to describe respondent's current health. As in HLY calculations [6], we considered any limitation as an indicator of disability. We imputed point estimates for respondents' sHLE based on the average of possible ages without limitation, depending on their coherent response patterns of current limitations, future ages with and without expected limitations and the expected age of death. Altogether, we defined five coherent response patterns (and corresponding imputation types) based on plausible orderings of current health with or without limitations, the onset of future limitations and the age of expected death (Section C in S1 Appendix). We considered response patterns as incoherent if at any future age expected limitations were followed by expected healthy states or if subjective life expectancy was lower

than current age. Incoherent respondents were excluded from the analysis. Examples for sLE imputation types and incongruent answer patterns are provided in Section D of S1 Appendix. The scatterplot of sHLE point estimates by age and imputation type are provided in the Figure of S1 Appendix. Throughout this paper the term sHLE will represent the subjective disability-free life expectancy from birth.

Abridged HLYs and actuarial life tables and for 2019 were provided by the HCSO. Both Eurostat and the HCSO calculates HLY's from GALI results of the EU-SILC survey using the Sullivan method, the only methodological difference being that the HCSO uses the mid-year population in the calculations, while the Eurostat takes into account the beginnings of the year [48]. Yearly life expectancy and mortality data were available for both sexes up to 100 age years [49].

We estimated actuarial HLYs for each age year from abridged HLY tables available in standard format 5-year age groups up to 84 years and a single 85+ year-old age group. The estimation process is detailed in the S2 Appendix. Estimating HLYs for ages over 85 years was beyond the scope of our study. From the differences of life expectancy and healthy life expectancy curves, we computed actuarial life years with disability (LYD). From the difference of sLE and sHLE we also computed subjective life-years with disability. Throughout this paper, sLYD will denote indirect estimates of subjective life-years with disability, as this measure were not directly queried in the survey.

## Sample survival curves

Subjective survival and subjective healthy survival functions were determined from sLE and sHLE via the Kaplan-Meier method.

In order to compare the subjective survival functions with actuarial survival projections for the sample, we constructed 20-year actuarial survival curves for 50–65 year-old male and female respondents via the cohort-compartment method (CCM) [50], using period life tables by $s_i = \frac{\sum_{x=50}^{65} N_x \prod_{i=0}^{20}(1-q_{x+i})}{N}$, where $s_i$ denotes the proportion of the sample for $i \in \{0,1,2,\ldots 20\}$ years on from current age $x \in \{50,51,\ldots 65\}$, $q_x$ is the conditional probability of death at age $x$, $N_x$ is the cohort sample size at age $x$ and $N$ is the total sample size. We obtained $q_x$ from the 2019 national life tables separately for males (S1 Table) and females (S2 Table).

We applied the following formula for the actuarial healthy survival projection for both sexes: $hs_i = \frac{\sum_{x=50}^{65} N_x p_x \prod_{i=0}^{20}(1-hq_{x+i})}{N}$, where $hs_i$ denotes the proportion of the sample surviving without limitations due to health problems for $i \in \{0,1,2,\ldots 20\}$ years on from current age $x \in \{50,51,\ldots 65\}$, $hq_x$ is the conditional probability of becoming limited at age $x$ given the respondent was healthy until age $x$, $N_x$ is the cohort sample size at age $x$, $p_x$ is the proportion without limitations at age $x$ and $N$ is the total sample size. The parameters are displayed in S1 Table for males and S2 Table for females.

## Explanatory variables

In addition to respondents' age and gender, we inquired about the longevity of close family members. Health status was recorded by the self-perceived health item of MEHM [10,11]. We asked whether respondents were recipients of any informal or formal care due to health problems or ageing. Lifestyle risks were captured via a self-perceived item of own lifestyle relative to others'. Also, smoking habits, frequency of alcohol intake, body mass index (calculated from reported weight and height) and weekdays with at least 10 minutes of uninterrupted sport / physical activity were recorded. Socioeconomic status was described by respondents' level of education, place of residence and net household income per capita according to the following:

net monthly household income was asked in 10 equal intervals (0–1720 USD) and an open-ended top category (>1720 USD). Category midpoints were transformed to a continuous variable using the method of Parker and Fenwick for the top category [51]. Household income was divided by the number of household members without adjustment for the number of children, and grouped according to the lowest, middle and uppermost national quintiles of per capita net monthly household income [52]. Data were collected in local currency and converted to USD using the average currency exchange rate of 2019 (1 USD = 290.65 HUF) [53]. We recorded the number of adults and <18-year-old children as proxy variables for respondents' social environment. Finally, we recorded the level of happiness on the 11-point numeric happiness scale. Higher scores indicated greater happiness [54].

## Statistical analysis

Sample characteristics and key exploratory variables were summarized via descriptive methods.

The current and future health expectation patterns of respondents ($H_x$, $H_{x+10}$, $H_{x+20}$) by age-group and gender were tabulated. $H_x$ represented current health, while $H_{x+10}$ and $H_{x+20}$ denoted health states in future time points with 10- and 20-year lag from the lower age-group boundary. For example, for the 60–69 years old age-group, $H_{x+10}$ denoted expected health at age 70, while $H_{x+20}$ denoted expected health at age 80. GALI categories were denoted as not limited, limited (but not severely) or severely limited. We denoted $H_{x+10}$ and $H_{x+20}$ as dead if sLE was shorter than the respective decade of $H_{x+10}$ or $H_{x+20}$. The proportion of respondents in future health states as well as the proportion transitioning between health states was compared via cross-tabulation and the Fischer's exact test.

We applied survival methods for the analysis of sLE and sHLE point estimates. Median sLE and median sHLE was calculated and subjective survival curves of the two sexes were compared via the log-rank test. Median and mean values of subjective and actuarial values were compared via the non-parametric sign-test and one-sample t-test, respectively.

We also compared sLE, LE as well as sHLE and HLE for age years directly via explorative graphical methods. To reduce the scatter of subjective estimates around the actuarial data, we applied a local polynomial smoothing for the subjective estimates (including a 95%CI band) using an Epanechikov kernel. The differences between subjective and actuarial values were compared to 0 by age-group and gender by the one-sample t-test. Furthermore, we explored the correlation structure of sLE and sHLE via scatterplots, including the lines connecting pairs of actuarial LE / HLE data points. Estimates of sLE, sHLE and sLYD were compared via the two-sample t-tests between subgroups.

For the multivariate modelling of sHLE, instead of using Type 5 imputed values, we applied interval-censored regression, where those with zero or negative remaining healthy time were considered to be interval-censored to (-∞,0), i.e., left censored at respondents' age (as for these subjects we did not have information when the limitation commenced), and non-censored for those where the remaining time was positive. A semiparametric proportional hazards model was used, with standard errors estimated through bootstrap [55]. sLE was modelled with multivariate Cox proportional hazards model. The model was stratified according to quantiles of age and close relatives' life span as these violated the proportionality assumption; the stratified model was acceptable, as evidenced by the test of Grambsch and Therneau [56]. Finally, sLYD was modelled with multivariate ordinary least squares (OLS) regression using robust standard errors.

Statistical analyses were conducted by Stata 14 statistical software [57] and R version 4.0 [58] using package icenReg version 2.0.15 [59].

## Results

### Sample

Basic sociodemographic characteristics of the survey sample (N = 1000), the study sample (N = 504) and the Hungarian general population [49] are displayed in Table 1. In the study sample, mean (±SD) age was 63 (±7.5) years. Male respondents, the 60–69 years old age-group and respondents with secondary and tertiary education levels were over-represented compared to the ≥50-year-old general population. Over third of respondents who answered the income item (34.9%, N = 148/424) were from the highest income group (5th quintile). Mean (±SD) happiness score was 6.7 (±2.3). According to GALI, 43.7% (N = 220/504) of respondents experienced any long-standing limitation due to health problems. The distribution of key explanatory variables is provided in Table 2. Smoking ($p = 0.026$), excessive alcohol intake ($p < 0.001$) and lack of exercise ($p = 0.005$) was more frequent among men than in women, while more women lived in single-adult households ($p < 0.001$).

**Table 1. Sample characteristics.**

| | | Study sample | | Population ≥50[a] | Survey sample | | Population ≥18[b] |
|---|---|---|---|---|---|---|---|
| | | N | % | % | N | % | % |
| Total | | 504 | 100 | - | 1000 | 100 | 100 |
| Gender | Male | 244 | 48.4 | 42.5 | 455 | 45.5 | 47.1 |
| | Female | 260 | 51.6 | 57.5 | 545 | 54.5 | 52.9 |
| Age-group | 18–29 | - | - | - | 101 | 10.1 | 17.2 |
| | 30–39 | - | - | - | 156 | 15.6 | 16.0 |
| | 40–49 | - | - | - | 201 | 20.1 | 19.6 |
| | 50–59 | 150 | 29.8 | 32.0 | 165 | 16.5 | 15.1 |
| | 60–69 | 261 | 51.8 | 34.5 | 275 | 27.5 | 16.3 |
| | ≥70 | 93 | 18.5 | 33.5 | 102 | 10.2 | 15.8 |
| Education | Primary | 152 | 30.2 | 55.0 | 300 | 30.0 | 45.4 |
| | Secondary | 196 | 38.9 | 27.5 | 422 | 42.2 | 33.3 |
| | Tertiary | 156 | 31.0 | 17.5 | 278 | 27.8 | 21.2 |
| Place of residence | Capital | 113 | 22.4 | 17.3 | 223 | 22.3 | 18.3 |
| | City/town | 268 | 53.2 | 53.2 | 523 | 52.3 | 52.4 |
| | Village | 123 | 24.4 | 29.5 | 254 | 25.4 | 29.3 |
| Region | Central Hungary | 171 | 33.9 | 29.2 | 338 | 33.8 | 31.0 |
| | Transdanubia | 149 | 29.6 | 31.1 | 280 | 28.0 | 30.2 |
| | Great Plain and North | 184 | 36.5 | 39.8 | 382 | 38.2 | 38.9 |
| Income | 1st quintile | 74 | 14.7 | 10.6 | 206 | 20.6 | 15.8 |
| | 2nd quintile | 75 | 14.9 | 19.2 | 149 | 14.9 | 20.1 |
| | 3rd quintile | 51 | 10.1 | 27.1 | 94 | 9.4 | 23.9 |
| | 4th quintile | 76 | 15.1 | 23.7 | 120 | 12.0 | 21.7 |
| | 5th quintile | 148 | 29.4 | 19.4 | 258 | 25.8 | 18.5 |
| | Missing | 80 | 15.9 | - | 173 | 17.3 | - |
| Inclusion criteria | Age ≥ 50 years | 504 | 100 | - | 542 | 54.2 | - |
| | Consistent responder | 504 | 100 | - | 914 | 91.4 | - |

[a]2019 Demographic Yearbook of Hungary [49], ≥50-year-old age group.

[b]2019 Demographic Yearbook of Hungary [49], distribution of age, gender and education are provided for the ≥18 age group, place of residence and region for the entire population of Hungary.

**Table 2. Distribution of key explanatory variables.**

| Variable | Level | Total | | Male | | Female | | Fischer exact[a] |
|---|---|---|---|---|---|---|---|---|
| | | N | % | N | % | N | % | p |
| GALI[b] | Severely limited | 29 | 5.8 | 18 | 7.4 | 11 | 4.2 | 0.340 |
| | Limited, but not severely | 191 | 37.9 | 91 | 37.3 | 100 | 38.5 | |
| | Not limited at all | 284 | 56.3 | 135 | 55.3 | 149 | 57.3 | |
| Close relatives' lifespan | 55–74 years | 148 | 29.4 | 77 | 31.6 | 71 | 27.3 | 0.524 |
| | 75–85 years | 222 | 44.0 | 102 | 41.8 | 120 | 46.2 | |
| | 85+ years | 134 | 26.6 | 65 | 26.6 | 69 | 26.5 | |
| Self-perceived health | Good (Very good / Good) | 234 | 46.4 | 113 | 46.3 | 121 | 46.5 | 0.515 |
| | Bad (Fair / Bad / Very Bad) | 270 | 53.6 | 131 | 53.7 | 139 | 53.5 | |
| Caregiver | Has caregiver | 58 | 11.5 | 27 | 11.1 | 31 | 11.9 | 0.960 |
| | No caregiver, but would need one | 19 | 3.8 | 9 | 3.7 | 10 | 3.8 | |
| | No caregiver, and do not need one | 427 | 84.7 | 208 | 85.2 | 219 | 84.2 | |
| Self-reported lifestyle | As healthy or healthier than others | 423 | 83.9 | 202 | 82.8 | 221 | 85.0 | 0.289 |
| | Less healthy than others | 81 | 16.1 | 42 | 17.2 | 39 | 15.0 | |
| Smoking | Non-smoker | 363 | 72.0 | 186 | 76.2 | 177 | 68.1 | 0.026 |
| | Current smoker | 141 | 28.0 | 58 | 23.8 | 83 | 31.9 | |
| Alcohol consumption | Max 3–4 days/week | 440 | 87.3 | 191 | 78.3 | 249 | 95.8 | <0.001 |
| | 5+days/week | 64 | 12.7 | 53 | 21.7 | 11 | 4.2 | |
| BMI[c] | <30 (Not obese) | 323 | 64.1 | 152 | 62.3 | 171 | 65.8 | 0.236 |
| | ≥30 (Obese) | 181 | 35.9 | 92 | 37.7 | 89 | 34.2 | |
| Physical activity | Some exercise (≥ 1 day per week) | 260 | 51.6 | 111 | 45.5 | 149 | 57.3 | 0.005 |
| | No exercise | 244 | 48.4 | 133 | 54.5 | 111 | 42.7 | |
| N of adults in household | Single adult | 160 | 31.7 | 53 | 21.7 | 107 | 41.2 | <0.001 |
| | Two or more adults | 344 | 68.3 | 191 | 78.3 | 153 | 58.8 | |
| N of children in household | No children | 459 | 91.1 | 224 | 91.8 | 235 | 90.4 | 0.345 |
| | One or more children | 45 | 8.9 | 20 | 8.2 | 25 | 9.6 | |

[a]Male vs Female
[b]GALI: Global Activity Limitation Indicator; [c]BMI: Body Mass Index.

## Future health expectations

The current self-perceived health status and subjective health expectation patterns by age-group and gender are shown in Fig 1. While the proportion of respondents without current limitation decreased significantly with age in men ($p = 0.007$), it was similar across age-groups in women ($p = 0.941$). Regarding future expectations, the proportion of healthy states (e.g., no limitations) decreased significantly with age ($p<0.001$) and time ($p<0.001$), while there was no difference between sexes ($p = 0.538$). However, in the 50–59 age-group, significantly more women than men expected limitations at the age of 70 ($p = 0.002$). While the proportion of expected deaths increased with age ($p<0.001$) and time ($p<0.001$), the difference between male and female respondents was not significant ($p = 0.404$). Severe limitations were expected infrequently (4.8%; 95%CI 3.5–6.3%) with increasing proportion over future time points ($p = 0.026$), but no significant differences between age-groups ($p = 0.922$) or sexes ($p = 0.183$). Moderate future limitations were expected by over one third of responders (36.0%, 95% CI 33.0–39.1%), with slight decrease over time ($p = 0.049$), a peak among the 60–69 years old age-group, and somewhat greater overall percentage in women (39.0%) than in men (32.8%) ($p = 0.042$).

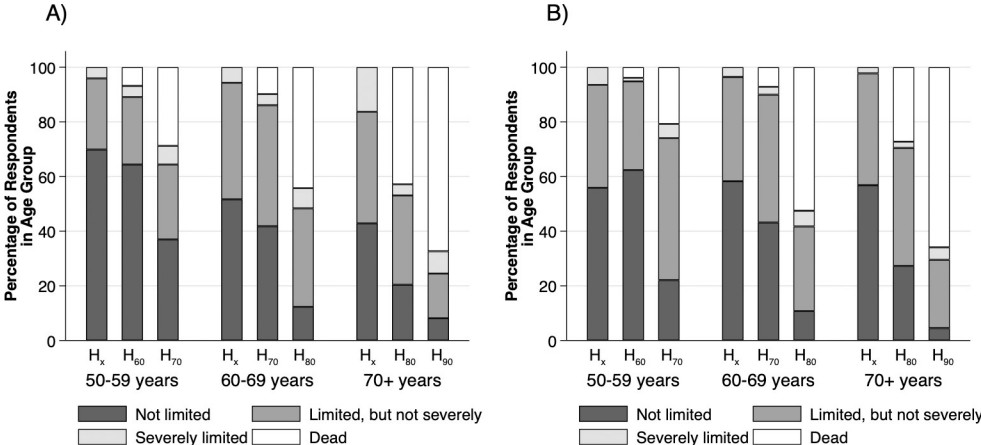

**Fig 1.** Current and future health expectation patterns by age group in A) men and B) women. $H_x$: current health status as measured by the Global Activity Limitations Indicator (GALI); $H_{60/70/80/90}$: subjectively expected health status on the GALI for ages 60/70/80/90 years.

The overall percentage of respondents transitioning from no limitations to expected moderate limitations between subsequent time points was 33.2% (95% CI 29.1–37.5%), with considerable acceleration rate between time points ($p < 0.001$) but no difference between age groups ($p = 0.398$) or sexes ($p = 0.075$). The percentage transitioning from no limitations to expected severe limitations was minimal (0.2%; 95%CI 0.0%-1.1%) with no difference between time points ($p = 0.555$), age groups ($p > 0.999$) or sexes ($p = 0.545$). However, the percentage transitioning from no limitation to death was 15.0% (95%CI 12.1–18.4%), with marked acceleration between time points ($p < 0.001$) and age groups ($p < 0.001$) and no sex differences ($p = 0.712$). From moderate limitations to severe limitations, the transition rate was 8.2% (95%CI 5.7–11.4%), with some increase over time ($p = 0.001$) but with no significant difference between ages ($p = 0.740$) or sexes ($p = 0.461$). However, the percentage transitioning from moderate limitations to death was 33.5% (95%CI 28.8–38.4%), with steep increase with time ($p < 0.001$) and age ($p < 0.001$) and no difference between sexes ($p > 0.999$). The transition from severe limitations to death was 51.1% (95%CI 35.8–66.3%), with no differences between groups.

## Comparisons of actuarial and subjective life expectancy and healthy life expectancy estimates

**Overall differences between actuarial and subjective life expectancy and healthy life expectancy estimates.** Point estimates for sLE clustered at 5-year multiples, and for sHLE at 64.75 and 74.75 years. The distributions of sLE and sHLE by gender are depicted in S1 Fig. For the total sample, median sHLE (66.8 years; 95%CI 65.8–68.3) was lower than the median HLE (72.7 years; 95%CI 72.2–72.7) by 5.9 years (two-sided sign-test, $p < 0.001$), and mean ($\pm$SD) sHLE (68.7$\pm$10.9 years) was lower by 4.0 years (two-sided paired t-test, $p < 0.001$) compared to HLE (72.8$\pm$3.5 years). There was no significant difference (two-sided sign test, $p = 0.142$) between median sLE (82 years; 95%CI 80–85 years) and actuarial LE (81.5 years; 95%CI 81.1–81.7 years). However, the 1.8 years difference between mean ($\pm$SD) sLE (82.8$\pm$9.6 years) and LE (81.1$\pm$3.0 years) was significant (two-sided paired t-test, $p < 0.001$). Median sLYD (12.25, 95%CI 10.25–14.25) was 4.3 years longer (two-sided sign-test, $p < 0.001$), than LYD (7.9 years, 95%CI 7.9–8.0) and mean ($\pm$SD) sLYD (14.1$\pm$4.4 years) exceeded LYD (8.3$\pm$0.9 years) by 5.8 years (two-sided paired t-test, $p < 0.001$).

**Actuarial and subjective survival and healthy survival by gender.** The actuarial and subjective survival and healthy survival functions for 50-65-year-olds are shown in Fig 2. For males, the actuarial survival estimate was below the lower boundary of the 95% CI range for nearly the entire 20-year period, while the subjective disability-free survival curves ran slightly above the actuarial estimate, within the 95% CI range. The differences were more nuanced for female respondents. While women overestimated both their overall and healthy survival in the forthcoming 10 years, the actuarial estimates fell within the 95% CI of the subjective survival curves between 10–20 years. The subjective survival curves of female respondents crossed the actuarial survival curves and run below the actuarial estimates on the 15 to 20-year year horizon.

For the entire sample, the difference between median sLE of men (82 years; 95%CI 80–85 years) and women (80 years; 95%CI 80–85 years) was not significant (log-rank test, *p* = 0.930), while median actuarial LE of males (79.1; 95%CI 78.7–79.5 years) and females (82.7; 95%CI

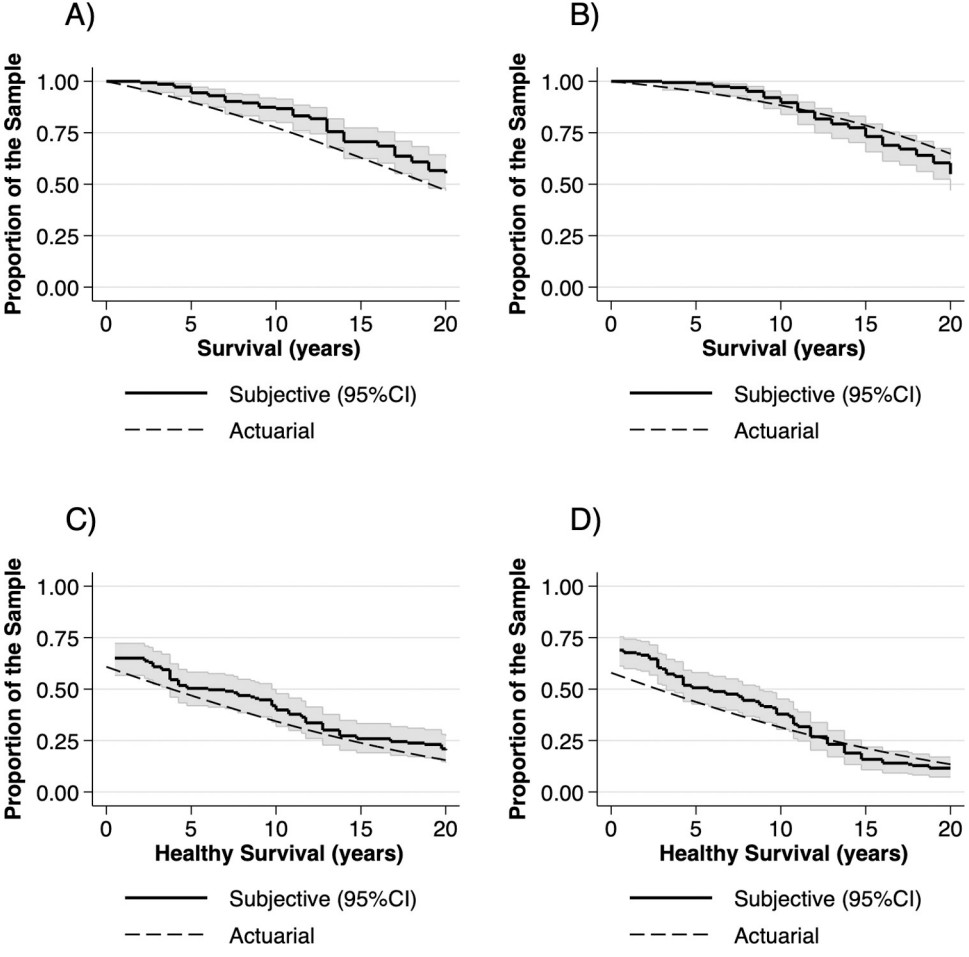

**Fig 2.** Comparison of 20-year actuarial and subjective curves of A) survival in men, B) survival in women, C) healthy survival in men and D) healthy survival in women in 50-65-year-olds. Actuarial survival curves were calculated for the sample age distribution via the cohort-compartment method, using conditional mortality data from the 2019 population life tables and estimated yearly conditional probabilities of becoming limited from the 2019 5-year-age-group healthy life expectancy (HLY) tables. Subjective survival curves were calculated from sample point estimates of subjective life expectancy (sLE) and imputed point-estimates of subjective healthy life expectancy (sHLE) from sLE point estimates and expected limitations at age 60,70 80 and 90 years measured by the adapted Global Activity Limitation Indicator (GALI) instrument.

82.4–82.9 years) differed significantly (log-rank test, $p<0.001$). Neither median sHLE (log-rank test, $p = 0.403$) nor median actuarial HLE (log-rank test, $p = 0.417$) differed between sexes.

**Actuarial vs subjective estimates by age group and gender.** sHLE, sLE and sLYD values were rather similar between the two sexes, except for lower sLYD in men than in women in the 50-59-year-old age group ($p = 0.002$) (Fig 3). However, the differences compared to actuarial estimates showed gender differences. sLE was overestimated by men from 60 years of age, and by women over 70 years of age (Fig 3A). The mean (±SD, $p$) difference between sLE and LE in the 50–59, 60–69 and ≥70-year-old age groups was 2.4 (±1.4, $p = 0.09$), 4.3 (±0.9, $p<0.001$) and 2.8 (±1.1, $p = 0.012$) years in men and -0.6 (±1.2, $p = 0.631$), -0.34(±0.66, $p<0.610$) and 3.4 (±1.1, $p = 0.002$) years in women, respectively. sHLE and HLE differed in men by -1.8 (±1.6, $p = 0.250$), -4.1 (±0.9, $p<0.001$) and -3.4 (±1.2, $p = 0.006$) years and by -6.7 (±1.2, $p<0.001$), -4.4 (±0.8, $p<0.001$) and -2.3 (±1.2, $p = 0.060$) years in women (Fig 3B). sLYDs were higher than LYDs by 4.2 (±1.0, $p<0.001$), 8.4 (±1.0, $p<0.001$) and 6.2 (±1.1, $p<0.001$) in men and 6.1 (±1.3, $p<0.001$), 4.1 (±0.8, $p<0.001$) and 5.6 (±1.1, $p<0.001$) years in women in the 50–59, 60–69 and ≥70-year-old age groups, respectively (Fig 3C).

**Differences between actuarial and subjective estimates relative to actuarial remaining time.** The respective differences between remaining sLE and LE, sHLE and HLE and sLYD

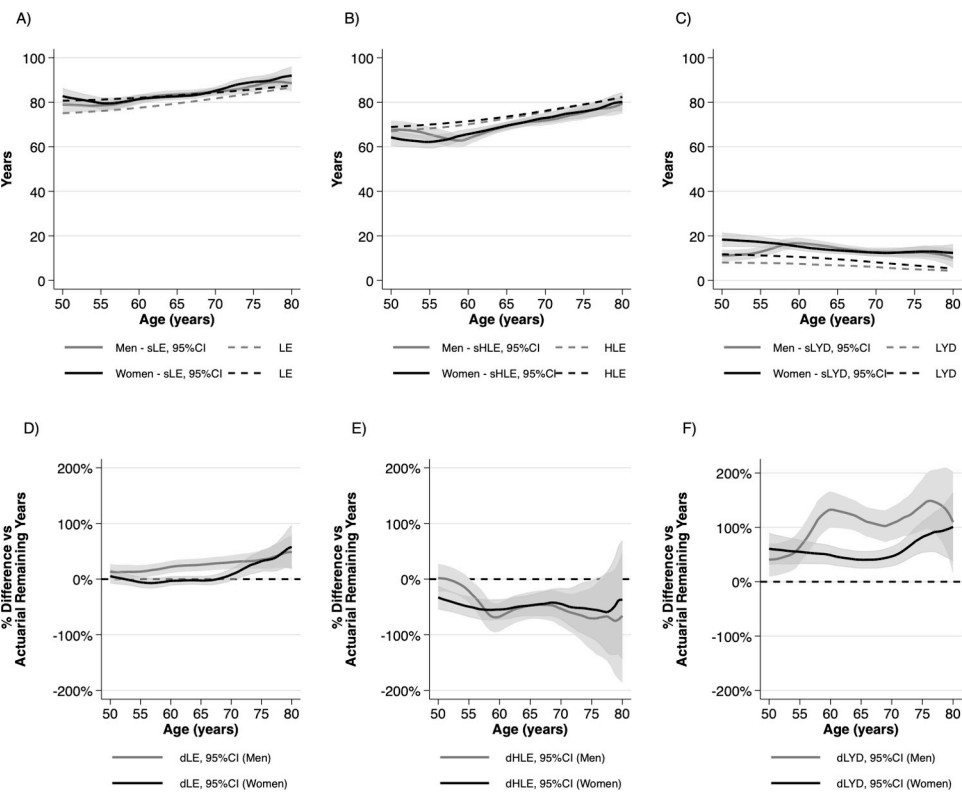

**Fig 3.** Actuarial vs subjective A) life expectancy, B) healthy life expectancy and C) life years with disability and difference between actuarial and subjective D) life expectancy, E) healthy life expectancy and F) life years with disability relative to remaining actuarial time by age and gender. LE: life expectancy, sLE: subjective life expectancy, HLE: healthy life expectancy, sHLE: subjective healthy life expectancy, LYD: life years with disability, sLYD: subjective life years with disability, dLE: difference of actuarial and subjective life expectancy relative to actuarial remaining life expectancy, dHLE: difference of actuarial and subjective healthy life expectancy relative to actuarial remaining healthy life expectancy; dLYD: difference of actuarial and subjective life years with disability relative to actuarial remaining life years with disability.

and LYD relative to the respondents' remaining actuarial LE, HLE and LYD are illustrated in Fig 3D (dLE), Fig 3E (dHLE) and Fig 3F (dLYD). While men overestimated remaining sLE on average by 23.3% ($p < 0.001$), estimates of women were accurate up to 70 years of age (mean difference: -2.3%, $p = 0.410$). ≥70-year-old women overestimated sLE by 31.1%, ($p < 0.001$). The overall difference between the accuracy of sLE estimates by men and women was significant ($p < 0.001$). Both men and women underestimated remaining sHLE. Although men were more accurate than women in the 50–59 age-group (-16.9% vs -22.7%, $p = 0.043$), the overall difference was not significant across all age groups (-48.6% vs -47.1%, $p = 0.885$). Both men and women overestimated their remaining sLYDs. Despite the difference was not significant in the 50–59 (53.6% vs 54.2%, $p = 0.972$) and ≥70-year-old age groups (121.1% vs 80.8%, $p = 0.130$), the overall difference between men's and women's remaining sLYD estimates was large and highly significant (101.6% vs 52.2%, $p < 0.001$).

**Association between subjective life expectancy and healthy life expectancy.** Corresponding sLE and sHLE values for each respondent were depicted on a scatterplot along with actuarial LE and HLE values (Fig 4). The Pearson correlation coefficient between sLE and sHLE in men and women was 0.60 and 0.48, respectively. We observed considerable individual variance in the difference between individual sLE and sHLE estimates (indirect sLYDs). According to the fitted local polynomial curves, on average, women's sLE estimates were close to actuarial LE, if their sHLE estimates were in the actuarial range. Even those women, who expected to become disabled early, expected to live on average up to 80 years. sLE was below 80 years for those ~50-year-old women, who were disabled at the time of survey and increased proportionally for those who expected long healthy lives. sLE estimates of men were greater than actuarial LE when sHLE was estimated in the actuarial range and as women, even with early disability expectations, men expected to live up to 80 years on average. However, sLE for ~50-year-old men who were disabled at the time of survey was below 70 years. Male and female scatterplots are shown overlaid in S2 Fig.

**Determinants of subjective estimates of life expectancy, healthy life expectancy and life years with disability.** Although sHLE, sLE and sLYD estimates of men and women were similar, the regression analyses revealed multiple differences between the two sexes in terms of the determinants of subjective expectations (Table 3). We interpreted the hazard ratios (HR)

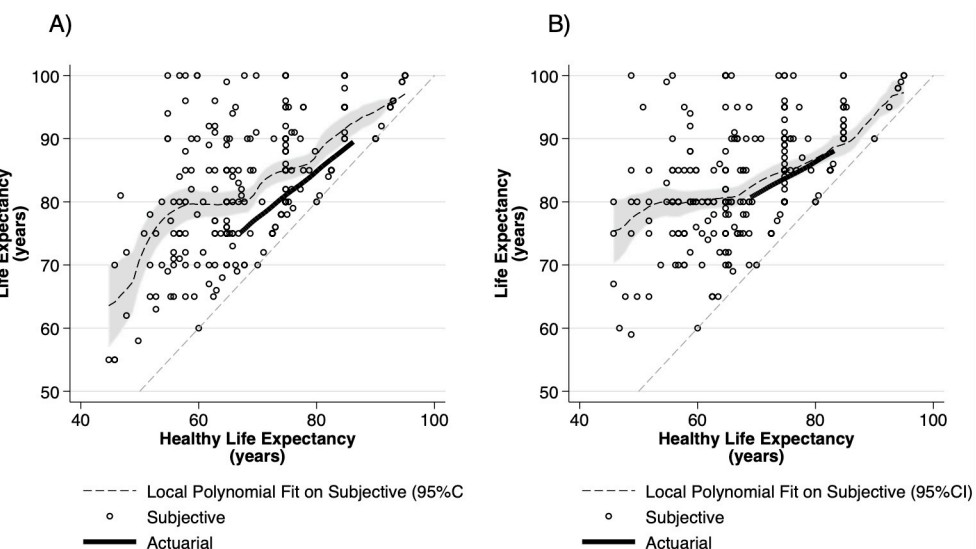

**Fig 4.** Healthy life expectancy vs life expectancy in A) wen and B) women.

in the survival models in terms of their effect on survival time: HR >1 suggested shorter, while HR<1 suggested longer sLE or sHLE versus baseline.

sHLE was mainly determined by self-perceived health in both men and women. Place of residence other than the capital was also associated with shorter sHLE estimates in both men

**Table 3. Regression analysis of subjective healthy life expectancy, subjective life expectancy and subjective life years with disability by gender.**

| | | sHLE[n] | | | | sLE[o] | | | | sLYD[p] | | | |
|---|---|---|---|---|---|---|---|---|---|---|---|---|---|
| | | Male | | Female | | Male | | Female | | Male | | Female | |
| | | HR[q] | p | HR | p | HR | p | HR | p | Beta | p | Beta | p |
| **Age** | Age (years) | 0.96* | 0.013 | 0.97 | 0.068 | - | - | - | - | -0.17 | 0.084 | -0.32** | 0.001 |
| **Close relatives' lifespan[a]** | 55–74 years | 1.37 | 0.247 | 1.12 | 0.638 | - | - | - | - | -0.76 | 0.607 | -0.80 | 0.631 |
| | ≥85 years | 0.77 | 0.387 | 0.98 | 0.940 | - | - | - | - | 1.12 | 0.506 | 5.21** | 0.004 |
| **Self-perceived health[b]** | Fair/Bad/Very Bad | 3.74** | <0.001 | 3.35** | <0.001 | 1.42 | 0.095 | 1.63** | 0.009 | 7.94** | <0.001 | 5.60** | 0.001 |
| **Caregiver[c]** | Has caregiver | 4.49 | 0.126 | 1.28 | 0.618 | 2.41** | 0.006 | 0.71 | 0.199 | -1.04 | 0.781 | 7.90** | 0.006 |
| | No caregiver, but needed | 5.38 | 0.537 | 0.54 | 0.441 | 2.86* | 0.034 | 1.53 | 0.364 | 2.14 | 0.551 | 5.42* | 0.047 |
| **Self-perceived lifestyle[d]** | Less healthy than others | 2.11 | 0.051 | 1.63 | 0.175 | 1.43 | 0.156 | 2.08** | 0.005 | 0.42 | 0.829 | -2.59 | 0.160 |
| **Smoking status[e]** | Current smoker | 1.20 | 0.572 | 1.18 | 0.424 | 1.07 | 0.737 | 1.20 | 0.329 | -0.19 | 0.917 | -0.25 | 0.870 |
| **BMI[f]** | ≥30 (Obese) | 0.87 | 0.581 | 0.91 | 0.677 | 0.56** | 0.003 | 1.19 | 0.316 | 3.71** | 0.009 | -1.11 | 0.428 |
| **Physical activity[g]** | No exercise | 0.88 | 0.571 | 0.74 | 0.186 | 0.98 | 0.912 | 0.84 | 0.326 | -1.93 | 0.138 | -0.76 | 0.582 |
| **Alcohol intake[h]** | ≥5 days/week | 0.82 | 0.470 | 0.43 | 0.262 | 0.83 | 0.397 | 1.28 | 0.599 | -1.15 | 0.478 | -5.50 | 0.081 |
| **Education[i]** | Primary | 0.74 | 0.353 | 0.88 | 0.628 | 1.14 | 0.520 | 0.92 | 0.696 | -1.55 | 0.329 | -0.35 | 0.833 |
| | Tertiary | 0.77 | 0.373 | 0.99 | 0.970 | 0.79 | 0.310 | 0.80 | 0.258 | 0.48 | 0.775 | 2.63 | 0.127 |
| **Place of residence[j]** | City/Town | 2.06* | 0.039 | 1.14 | 0.558 | 1.26 | 0.307 | 1.12 | 0.570 | 1.60 | 0.398 | 0.87 | 0.567 |
| | Village | 2.10 | 0.060 | 2.05** | 0.009 | 1.00 | 0.996 | 0.81 | 0.370 | 3.86 | 0.066 | 2.98 | 0.135 |
| **Income[k]** | Q1 | 0.85 | 0.661 | 0.90 | 0.710 | 1.48 | 0.110 | 1.03 | 0.913 | -2.72 | 0.166 | -0.16 | 0.937 |
| | Q5 | 1.21 | 0.450 | 0.77 | 0.236 | 1.22 | 0.328 | 0.89 | 0.557 | -1.58 | 0.328 | -2.37 | 0.128 |
| **Household** | One or more adults[l] | 1.40 | 0.233 | 1.38 | 0.138 | 1.30 | 0.244 | 1.20 | 0.301 | -0.16 | 0.931 | 0.62 | 0.702 |
| | One or more children[m] | 2.48 | 0.069 | 0.97 | 0.925 | 0.52 | 0.050 | 1.00 | 0.991 | 6.25* | 0.023 | 1.09 | 0.594 |
| **Happiness** | Happiness (0–10) | 0.97 | 0.557 | 0.91 | 0.071 | 0.91* | 0.032 | 0.88** | 0.004 | 0.33 | 0.297 | 0.15 | 0.649 |
| **Intercept** | | - | - | - | - | - | - | - | - | 15.50 | 0.020 | 23.86 | 0.002 |
| **N** | | 204 | | 220 | | 204 | | 220 | | 204 | | 220 | |

* p>0.05

**p<0.01.

[a]base:75–84 years

[b]base: Very good/Good

[c]base: No, and not needed

[d]base: Healthier or as healthy as others

[e]base: Never smoked or quitted

[f]base: <30

[g]base: Some exercise

[h]base: Lifetime abstinence to max 4 occasions/week

[i]base: Secondary

[j]base: Capital

[k]base: Q2-Q4

[l]base: Single adult

[m]base: None

[n]sHLE: subjective healthy life expectancy

[o]sLE: subjective life expectancy

[p]sLYD: subjective life years with disability

[q]HR: hazard ratio.

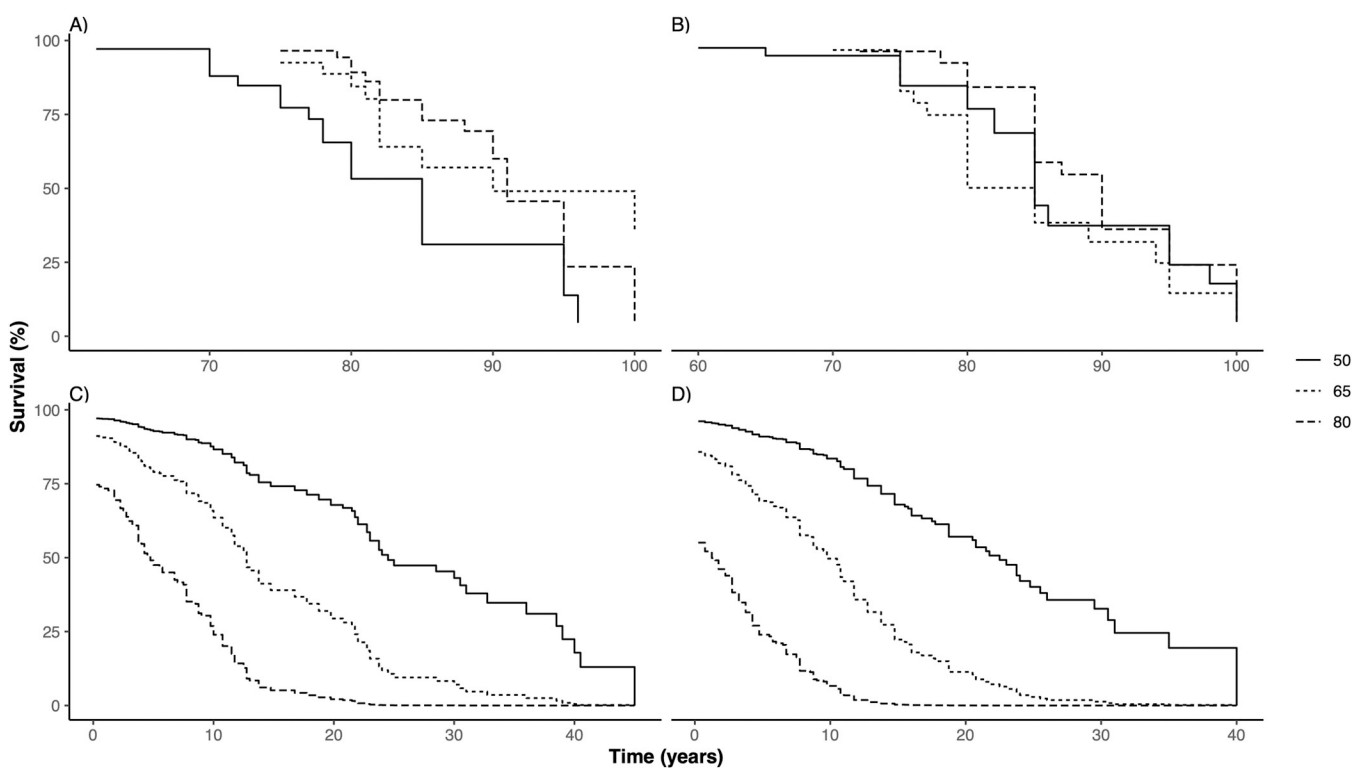

**Fig 5. Adjusted survival curves illustrating the effect of age on subjective life expectancy (sLE) and healthy life expectancy (sHLE) by gender sLE: Subjective life expectancy, sHLE: Subjective healthy life expectancy.**

and women, although with differences between the coefficients of cities/towns and villages. While older age was associated with longer sHLE in men, the effect of age was not significant in women.

Wald tests comparing the lowest and highest categories suggested that longer lifespan of close relatives was associated with longer sLE in both men ($p = 0.009$) and women ($p<0.001$). In order to meet the proportional hazards assumption, the final sLE model was stratified by age groups and close relatives' lifespan, so these variables were not included as predictors in the final sLE model. While greater happiness was associated with longer sLE in both men and women, the effect of health status and lifestyle was different between the two sexes. Low self-perceived health was associated with shorter sLE in women, but the association was not significant in men. However, having or needing a caregiver decreased significantly men's but not women's sLE. Less healthy self-perceived lifestyle compared to others was associated with significantly shorter sLE only in women, while obesity was associated with longer sLE in men.

Although low self-perceived health was a predictor of longer sLYD estimates in both men and women, other predictors were different between the two sexes. While higher age decreased the sLYD expectations of women, its effect was not significant in men. Long lifespan of close relatives and having or needed a caregiver were associated with longer sLYD estimates in women but on in men. On the other hand, in men, obesity was associated with longer sLYD estimates as well as living in a household with children.

Usual determinants of morbidity and mortality, such as education, income or lifestyle related factors, such as smoking, excessive drinking or lack of exercise were not associated with either sHLE, sLE or sLYD estimates. The adjusted survival curves illustrate the effect of age (Fig 5) and key predictor variables (Fig 6) on sLE and sHLE in both sexes.

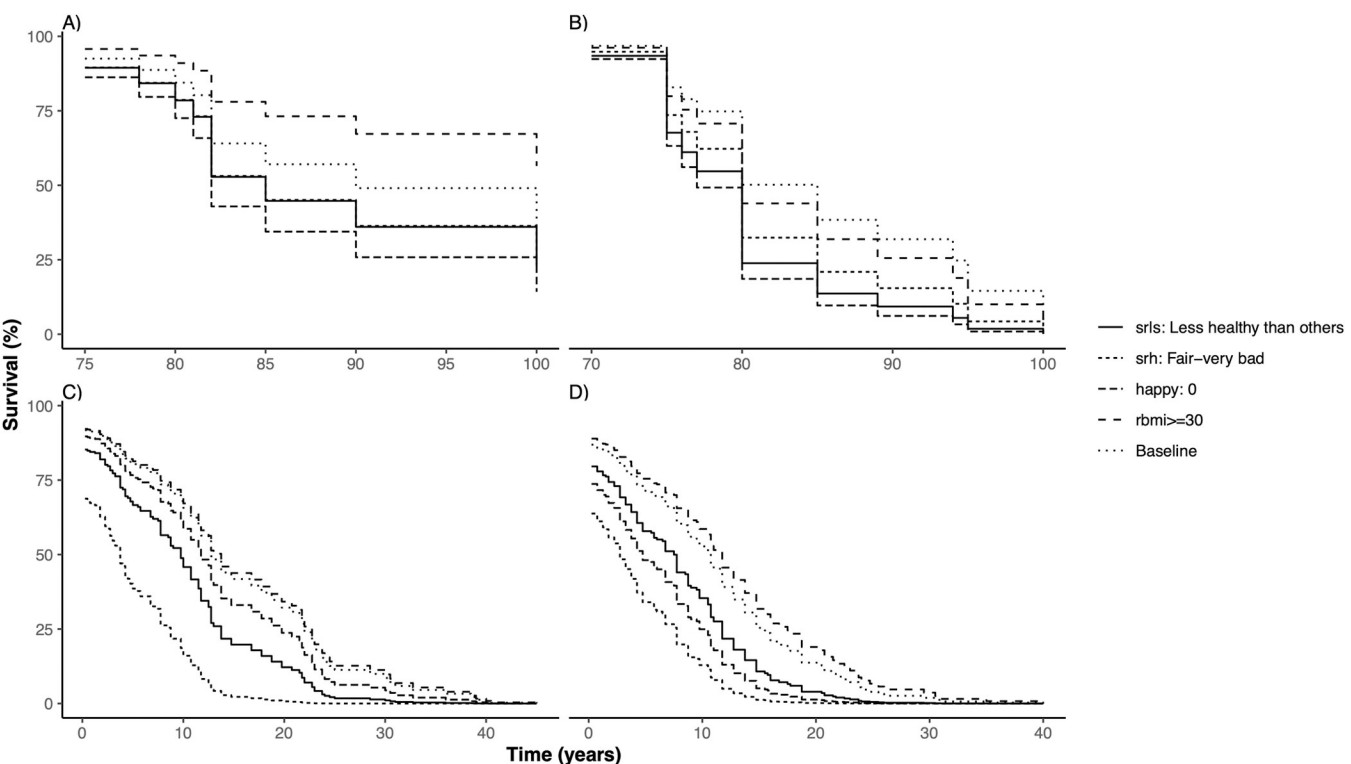

**Fig 6. Adjusted survival curves illustrating the effect of age key predictors on life expectancy (sLE) and healthy life expectancy (sHLE) by gender sLE: Subjective life expectancy, sHLE: Subjective healthy life expectancy.**

## Discussion

We explored subjective health expectations of the ≥50-year-old online general population via a cross-sectional online survey. According to our best knowledge, this is the first study that explores sHLE using the adapted GALI instrument, and thereby, provides comparable sHLE estimates to HLYs reported by Eurostat [6]. We estimated sHLE from self-reported current long-standing limitations due to health problems, expected limitations at ages 60, 70, 80 and 90 as well as point-estimates for subjective life-expectancy. While estimates of sLE and LE were rather similar, median sHLE was 5.9 years shorter than median HLE. Despite the greater median actuarial LE of women compared to men, we found no gender differences between the median sLE, HLE and sHLE values. sHLE was mainly determined by self-perceived health status and place of residence. The predictors of sHLE and sLE were different, and gender differences in determinants of sLE were more apparent compared to those of sHLE.

In the 50–59 age group, men's sHLE estimates were closer to actuarial HLE than women's, while men and women underestimated sHLE to a similar extent from age 60. Previous studies of subjective health expectations based on the EQ-5D-3L instrument in the Netherlands also demonstrated that people underestimate future health beyond 70 years of age [41].

Similarly to findings in the US, men overestimated their sLE [22]. In our study sLYD was also overestimated by men. Women estimated accurately sLE up to age 70, while overestimated sLYD across all ages and sLE in the ≥70-year-old age group. Altogether, men from 60 and women from 70 years of age expected to succumb earlier to disability, but withstand it longer compared to their actuarial estimates. The subjective LE and LYD estimates of women were closer to actuarial estimates than those of men.

The individual variability of sHLE and sLE estimates was great among respondents. Among the well-established determinants of sLE such as the longevity of forebears [14–18] self-perceived health status [16,18–20], age [16,18,19], gender [16,18,21–23], lifestyle-related risks [18,24–26], socio-economic status [21,27] and psychosocial factors [18,20,28–31], we found association of sHLE with self-perceived health, age while self-perceived health, close relatives' longevity, social conditions, happiness and perceived lifestyle influenced sLE.

In our sample, self-perceived health was the strongest predictor of sHLE, which is an important similarity with previous studies. In the Dutch population age, gender, current health status, perceived lifestyle and close relatives' lifespan were significant determinants of future health [41,42]. Furthermore, in addition to age, current health status, perceived lifestyle and close relatives' lifespan, education, employment and income were significant predictors of future health (as measured with the EQ-5D-3L) in the Hungarian general population indicating broad similarities between countries with different health status, health systems and economic level [60]. In studies with chronic patients, treatment status and informal care also influenced future health expectations [43,44]. Living in country towns or villages was an indicator of shorter sHLE in our sample, while place of residence was not considered in previous studies [41,42,60]. The association of sHLE and the place of residence deserves further investigation in the light of the considerable territorial health inequalities of Hungary. Compared to the affluent districts of the capital, life expectancy at birth may be up to 11 years lower in rural regions [61].

Important determinants of health, such as education, income, and lifestyle-related factors (smoking, physical activity, alcohol intake and obesity) did not affect sHLE in our sample, while in the Dutch population shorter future health expectations were associated with the presence of objective lifestyle risks, such as smoking, lack of exercise or unhealthy diet [41,42]. Our results were indicative of the population's ignorance about the negative consequences of unhealthy lifestyles, which may contribute to the unfavourable ranking of Hungary in Europe in terms of lifestyle risks and potentially preventable mortality [62]. The association of obesity with increased life expectancy in men is particularly alarming, and the association of sLE in women with self-perceived lifestyle but not with established risk factors may indicate the presence of health-related misbeliefs in the general population. These findings support the need for reinforced health education and health promotion activities in Hungary [63].

Our study revealed gender differences in future expectations about health, disability and longevity. In case of sHLE, the gender differences in terms of age and place of residence were subtle. Yet, we observed considerable differences between men and women in terms of how the social environment influenced sLE. Altogether, higher age was associated with shorter sHLE in men, but with decreased of sLYD in women. Becoming dependent on a caregiver was associated with the expectation of earlier death (shorter sLE) in men, but with extended survival of disability (longer sLYD) in women. Those men, who lived in a household with children, expected to live longer with disabilities, resembling the pattern of female respondents. Self-perceived health was associated with sLE in women but not in men. Also, the longevity of close relatives was associated with sLYD in women, but not in men. Other studies also revealed gender differences in subjective health expectations [41–43,60]. Further studies of the gender-related differences in health expectations may improve our understanding about gender differences in health behaviours [64], or retirement and financial planning [65].

It has to be noted, that methodological differences may hamper direct comparison of our results with previous studies of subjective health expectations. Since GALI measures limitations due to health problems, while EQ-5D-3L is a measure of health-related quality of life, the two instruments may reflect different aspects of future health expectations, which is worthwhile to explore in greater depth. We are aware of a single study that measured subjective

health expectations in terms of limitations on a 10-year horizon. The sample comprised the 40-55-year-old population [66]. Our study was conducted in the ≥50-year-old population, while other studies of future health also involved younger respondents from 18 years of age [41,42,60]. Due to the selection bias of survivors and the differences in timeframe for which the estimation is performed (i.e., regardless of age, all respondents estimated their health status for the same fixed ages), sHLE estimates can be particularly sensitive to the age distribution of the sample.

In our study, sHLE estimates were based on expectations of any future limitations. However, 4.8% of respondents expected severe limitations at some time-point. In the 2019 EU-SILC survey 17.3% of the 65+ years old Hungarian population reported severe limitations, while the range of country reports spanned between 6.5% and 25.9% [67]. While the difference may reflect the general sampling bias of online surveys [68], expectations about severe health states and death may indicate the health preferences of the general population [69] as well as how health systems can cope with the growing burden of serious health-related suffering [70,71].

The strength of our study is that our GALI-based sHLE estimates are comparable with Eurostat's HLY or DFLE estimates. Furthermore, the sample size allowed the inclusion of a wide range of predictor variables in separate models for men and women, thereby allowing the explore the differences between the two sexes in great detail.

However, some limitations have to be mentioned. While sLE was based on point-estimates, and yearly mortality data were available for comparison with actuarial LE values, sHLE was estimated from expected GALI responses at four time points, current GALI and sLE values involving a number of data imputation steps. The start date of existing limitations was not available. Furthermore, yearly actuarial HLE values were estimated from 5-year HLY tables. Although we employed reasonable data transformations, some of our assumptions, such as the applying the same imputation rule for the start-date of existing limitations across all ages may have influenced results. However, by applying interval regression during the survival analysis of left-censored sHLE data, this effect was somewhat mitigated. Furthermore, we included the ≥50-year-old online population in our study and limited the scope of some analyses to 20-years. Therefore, subjective healthy life expectations of younger generations, and of those who are outside the reach of online surveys have remained unexplored.

## Conclusions

In order to maintain the sustainability of healthcare and social systems under the pressure of growing expenditures and aging populations in Europe, is increasingly important to gain deeper understanding of people's underlying subjective perceptions and expectations about their future health and longevity beyond the dynamics of objective demographic indicators.

We demonstrated that in addition to the already established sLE, sHLE may be a feasible indicator of individual health expectations. The study of sHLE may open new avenues for interdisciplinary collaboration between demographers and scientists from the field of public health, health psychology or health economics. According to our results, in the ≥50-year-old population, median subjective healthy life span was lower by 5.9 years compared to actuarial HLE estimates. The similar sHLE estimates of men and women did not reflect the gender differences in LE. sHLE and sLE were determined by different factors.

In Hungary, the association between geographical health inequalities, regional differences in sHLE and their cultural and structural determinants deserve more exploration. Furthermore, the lack of association between sHLE and objective lifestyle-related risk factors, such as smoking status, physical exercise, obesity or alcohol intake were an alarming finding, which may partly be related to the high prevalence of lifestyle related risks among the Hungarian

general population. The determinants of subjective health expectancy and its association with actual health behaviours as well as health-related or economic decisions warrants further investigation.

## Supporting information

**S1 Fig.** Distribution of A) subjective life expectancy (sLE) in men, B) sLE in women, C) subjective healthy life expectancy (sHLE) in men and D) sHLE in women.
(PDF)

**S2 Fig. Subjective life expectancy (sLE) versus subjective healthy life expectancy (sHLE) in men and women.**
(PDF)

**S1 Table. Actuarial healthy life expectancy estimates for males by age years.**
(DOCX)

**S2 Table. Actuarial healthy life expectancy estimates for males by age years.**
(DOCX)

**S1 Appendix. Details of survey items on subjective expectations and applied data transformations.**
(PDF)

**S2 Appendix. Estimating actuarial healthy life years (HLY) for each age year from abridged HLY tables.**
(PDF)

**S1 File.**
(XLSX)

## Acknowledgments

The authors thank Dr. László Tóthfalusi and Dr. Miklós Farkas for their useful comments.

## Author Contributions

**Conceptualization:** Zsombor Zrubka, Áron Kincses, Levente Kovács, László Gulácsi, Márta Péntek.

**Data curation:** Áron Kincses.

**Formal analysis:** Zsombor Zrubka, Tamás Ferenci.

**Investigation:** Áron Kincses, László Gulácsi, Márta Péntek.

**Methodology:** Zsombor Zrubka, Áron Kincses, Tamás Ferenci, Márta Péntek.

**Resources:** Zsombor Zrubka, Levente Kovács, László Gulácsi.

**Supervision:** Áron Kincses, Levente Kovács, László Gulácsi, Márta Péntek.

**Validation:** Áron Kincses.

**Visualization:** Zsombor Zrubka, Tamás Ferenci.

**Writing – original draft:** Zsombor Zrubka.

**Writing – review & editing:** Zsombor Zrubka, Áron Kincses, Tamás Ferenci, Levente Kovács, László Gulácsi, Márta Péntek.

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
