## [Decision Letter · Decision Letter 0]

6 Dec 2021

PONE-D-21-03658Comparing Actuarial and Subjective Healthy Life Expectancy Estimates: a Cross-Sectional Survey Among the General Population in HungaryPLOS ONE

Dear Dr. Zrubka,

Thank you for submitting your manuscript to PLOS ONE. After careful consideration, we feel that it has merit but does not fully meet PLOS ONE’s publication criteria as it currently stands. Therefore, we invite you to submit a revised version.

We look forward to receiving your revised manuscript.

Kind regards,

Rashidul Alam Mahumud, MPH, MSc, PhD

Academic Editor

PLOS ONE

Journal Requirements:

2. Please upload a copy of Figure 5 & 6, to which you refer in your text on page 26 & 27. If the figure is no longer to be included as part of the submission please remove all reference to it within the text.

Reviewers' comments:

Reviewer's Responses to Questions

**Comments to the Author**

1. Is the manuscript technically sound, and do the data support the conclusions?

Reviewer #1: Yes

2. Has the statistical analysis been performed appropriately and rigorously? 

Reviewer #1: Yes

3. Have the authors made all data underlying the findings in their manuscript fully available?

Reviewer #1: Yes

4. Is the manuscript presented in an intelligible fashion and written in standard English?

Reviewer #1: No

5. Review Comments to the Author

Reviewer #1: Overall

Interesting paper and authors did hard work on it. However, it needs extensive revision to make publishable. It looks like directly copied from the report and too long which need entire revision.

The general comments are as follows:

Abstract

Methods line 29; GALI? write full form that appears first time.

Results line 32: 504? Replace it with the word.

Introduction

Page 3: line 56-57 (last sentence); looks like unnecessary.

Page 4: line 59; What are multiple methods? It might be better to introduce the paragraph with multiple methods and start describing them.

Page 4: line 68-83; this looks to long description, please merge the important things in a single sentence.

Page 5-6: Line 108-104; not necessary such details you can finish in a single sentence

Methods

Page 6: line 121; first sentence not needed.

Page 6: line 122 and 126; you already mentioned >50 in the objective; Why you collect 18-65 years? Make it clear.

Page 6: Please write sample and design section clearly and sequentially by referring one published article from the journal. It looks like mixed everything here.

Page 7: line 139-143; I suggest not to write the whole questions and answer here. (E.g. The opinion question was formulated …… and response was generated ……..)

Page 7-10: Line 132-201; I don’t think we need this much long explanation here.

Page 11-12: Line 221-247; explanatory variables section; please rewrite this no need to describe each questions in that details.

Page 20: line 376; Full form of GALI is already mentioned in the previous section so write short form GALI only.

Page 32: line 612: Can you make clear it clear?

Page 32: line 614: What is the reason behind ‘need of more explanation’?

6. PLOS authors have the option to publish the peer review history of their article (what does this mean?). If published, this will include your full peer review and any attached files.

Reviewer #1: No

---

## [Author Response · Author response to Decision Letter 0]

22 Jan 2022

Response to Reviewers

Title: Comparing Actuarial and Subjective Healthy Life Expectancy Estimates: a Cross-Sectional Survey Among the General Population in Hungary

Manuscript ID: PONE-D-21-03658

Dear Editors, Dear Reviewer, 

Thank you very much for reviewing our manuscript. We supplied the missing figures, updated supplementary files and revised file-naming conventions as per the journal authors’ guide. Our responses to reviewers’ comments are detailed below. 

Jan 20th, 2022

The Authors

Reviewer #1: Overall

Interesting paper and authors did hard work on it. However, it needs extensive revision to make publishable. It looks like directly copied from the report and too long which need entire revision.

[Authors] Thank you for the suggestions about making our manuscript more accessible by readers. We have abbreviated and simplified the entire paper and moved several technical details to supplementary files. 

The general comments are as follows:

Abstract

Methods line 29; GALI? write full form that appears first time.

[Authors] Thank you for pointing out, we introduced the acronym in the abstract (see Revised Manuscript line 12)

Results line 32: 504? Replace it with the word.

[Authors] Thank you for the suggestion, we replaced to „five hundred and five”. (see Revised Manuscript line 16)

Introduction

Page 3: line 56-57 (last sentence); looks like unnecessary.

[Authors] Thank you for the suggestion, we shortened the sentence. (see Revised Manuscript lines 37-40) 

Page 4: line 59; What are multiple methods? It might be better to introduce the paragraph with multiple methods and start describing them.

[Authors] Thank you for the suggestion, we re-edited and shortened this section. (see Revised Manuscript line 41)

Page 4: line 68-83; this looks to long description, please merge the important things in a single sentence.

[Authors] Thank you for the suggestion, we re-edited and shortened this section. (see Revised Manuscript lines 41-51)

Page 5-6: Line 108-104; not necessary such details you can finish in a single sentence

[Authors] Thank you for the suggestion, we re-edited and shortened this section. (see Revised Manuscript lines 70-73)

Methods

Page 6: line 121; first sentence not needed.

[Authors] Thank you for the suggestion, we re-edited and clarified this section. (see Revised Manuscript line 80)

Page 6: line 122 and 126; you already mentioned >50 in the objective; Why you collect 18-65 years? Make it clear.

[Authors] Thank you for the suggestion, we re-edited and clarified this section. The analysis was conducted on a sub-sample of a larger survey on health- and longevity-related expectations. While subjective life expectancy was inquired on the 18–65-year-old sample via a single point estimate (expected age of death), healthy life expectancy was computed from disability expectations for 60,70,80 and 90 years of age. To limit the inflation of measurement error for those respondents, who expected the onset of disability before age 60, we limited the study of healthy life expectancy to individuals who were 50+ years old. (e.g., healthy at age 18 vs age 50, if both expect disability by age 60). We added the following sentence: 

„Those respondents were selected in the study sample, who were ≥50-year-old and provided coherent answers to questions related to future health expectations from 60 years of age. Younger individuals were excluded to avoid inflated measurement error when the onset of expected disability was �60 years.” 

(see Revised Manuscript lines 88-91)

Page 6: Please write sample and design section clearly and sequentially by referring one published article from the journal. It looks like mixed everything here.

[Authors] Thank you for the suggestion, we re-edited this section. We hope that the sequence of reported information provides more clarity about our survey. (see Revised Manuscript lines 80-91)

Page 7: line 139-143; I suggest not to write the whole questions and answer here. (E.g. The opinion question was formulated …… and response was generated ……..)

Page 7-10: Line 132-201; I don’t think we need this much long explanation here.

[Authors] Thank you for the suggestion. We abbreviated the methods section and moved most of the technical details to supplementary files. (see Revised Manuscript lines 94-129)

Page 11-12: Line 221-247; explanatory variables section; please rewrite this no need to describe each questions in that details.

[Authors] Thank you for the suggestion. We abbreviated this section. (see Revised Manuscript lines 148-166)

Page 20: line 376; Full form of GALI is already mentioned in the previous section so write short form GALI only.

[Authors] Thank you for noting. We suggest keeping this caption text to Fig. 2 unchanged, so readers can interpret all figures without looking up information elsewhere in the manuscript. (see Revised Manuscript lines 287-296)

Page 32: line 612: Can you make clear it clear?

[Authors] Thank you for noting. We corrected the sentence as follows: „sHLE and sLE were determined by different factors.” (see Revised Manuscript line 532)

Page 32: line 614: What is the reason behind ‘need of more explanation’?

[Authors] Thank you for the question. There are considerable geographical inequalities of health in Hungary. One may hypothesize that low health expectations are determined culturally and contribute to the unhealthy lifestyles and overall inferior health of rural populations. However, low health expectations may as well be determined by structural factors, such as inferior health infrastructure and low access to healthcare, unemployment and poverty in rural Hungary. Therefore, we amended this sentence as follows: 

“In Hungary, the association between geographical health inequalities, regional differences in sHLE and their cultural and structural determinants deserve more exploration.”

(see Revised Manuscript line 533-534)

---

## [Decision Letter · Decision Letter 1]

16 Feb 2022

Comparing Actuarial and Subjective Healthy Life Expectancy Estimates: a Cross-Sectional Survey Among the General Population in Hungary

PONE-D-21-03658R1

Dear Dr. Zrubka,

We’re pleased to inform you that your manuscript has been judged scientifically suitable for publication and will be formally accepted for publication once it meets all outstanding technical requirements.

Kind regards,

Rashidul Alam Mahumud, MPH, MSc, PhD

Academic Editor

PLOS ONE

Additional Editor Comments (optional):

Reviewers' comments:

Reviewer's Responses to Questions

**Comments to the Author**

1. If the authors have adequately addressed your comments raised in a previous round of review and you feel that this manuscript is now acceptable for publication, you may indicate that here to bypass the “Comments to the Author” section, enter your conflict of interest statement in the “Confidential to Editor” section, and submit your "Accept" recommendation.

Reviewer #1: All comments have been addressed

2. Is the manuscript technically sound, and do the data support the conclusions?

Reviewer #1: Yes

3. Has the statistical analysis been performed appropriately and rigorously? 

Reviewer #1: Yes

4. Have the authors made all data underlying the findings in their manuscript fully available?

Reviewer #1: Yes

5. Is the manuscript presented in an intelligible fashion and written in standard English?

Reviewer #1: Yes

6. Review Comments to the Author

Reviewer #1: (No Response)

7. PLOS authors have the option to publish the peer review history of their article (what does this mean?). If published, this will include your full peer review and any attached files.

Reviewer #1: No

---

## [Editor Report · Acceptance letter]

2 Mar 2022

PONE-D-21-03658R1 

Comparing Actuarial and Subjective Healthy Life Expectancy Estimates: a Cross-Sectional Survey Among the General Population in Hungary 

Dear Dr. Zrubka:

I'm pleased to inform you that your manuscript has been deemed suitable for publication in PLOS ONE. Congratulations! Your manuscript is now with our production department. 

Kind regards, 

on behalf of

Dr. Rashidul Alam Mahumud 

Academic Editor

PLOS ONE